# Evaluating the Impact of Modic Changes on Operative Treatment in the Cervical and Lumbar Spine: A Systematic Review and Meta-Analysis

**DOI:** 10.3390/ijerph191610158

**Published:** 2022-08-16

**Authors:** Mark J. Lambrechts, Parker Brush, Tariq Z. Issa, Gregory R. Toci, Jeremy C. Heard, Amit Syal, Meghan M. Schilken, Jose A. Canseco, Christopher K. Kepler, Alexander R. Vaccaro

**Affiliations:** Rothman Orthopedic Institute, Thomas Jefferson University, Philadelphia, PA 19107, USA

**Keywords:** Modic changes, lumbar spine, cervical spine, patient reported outcomes, anterior cervical discectomy and fusion, lumbar fusion, discectomy

## Abstract

Modic changes (MCs) are believed to be potential pain generators in the lumbar and cervical spine, but it is currently unclear if their presence affects postsurgical outcomes. We performed a systematic review in accordance with the Preferred Reporting Items for Systematic Reviews and Meta-Analyses (PRISMA) guidelines. All studies evaluating cervical or lumbar spine postsurgical outcomes in patients with documented preoperative MCs were included. A total of 29 studies and 6013 patients with 2688 of those patients having preoperative MCs were included. Eight included studies evaluated cervical spine surgery, eleven evaluated lumbar discectomies, nine studied lumbar fusion surgery, and three assessed lumbar disc replacements. The presence of cervical MCs did not impact the clinical outcomes in the cervical spine procedures. Moreover, most studies found that MCs did not significantly impact the clinical outcomes following lumbar fusion, lumbar discectomy, or lumbar disc replacement. A meta-analysis of the relevant data found no significant association between MCs and VAS back pain or ODI following lumbar discectomy. Similarly, there was no association between MCs and JOA or neck pain following ACDF procedures. Patients with MC experienced statistically significant improvements following lumbar or cervical spine surgery. The postoperative improvements were similar to patients without MCs in the cervical and lumbar spine.

## 1. Introduction

Vertebral bone marrow edema is a recognized clinical entity that is believed to be associated with degenerative disc disease and back pain [1]. The aberrant signal changes identified with magnetic resonance imaging (MRI) were characterized into specific types by Modic et al., and were thus termed Modic changes (MC). Each distinct signaling phenotype is characterized by their unique signal intensity in MRI. Type I changes are characterized by hypointense T1- and hyperintense T2-weighted images, type II changes are recognized as hyperintense signaling on T1- and isointense to slightly hyperintense signaling on T2-weighted imaging, while type III changes are predominantly hypointense on the T1 and T2-weighted images [1]. Histological analysis has uncovered that the MRI signal intensity corresponds to anatomical changes. For example, type I MCs demonstrate bone marrow replacement with fibrovascular stroma, type II MCs are associated with fatty infiltration of the bone marrow, and type III MCs are characterized by sclerotic endplate changes [2,3]. Longitudinal studies have suggested that these changes exist on a continuum with transformations reported between all types, but most commonly with the progression of type I to type II [2,4,5,6,7,8]. Degenerative disc disease is the most frequently identified disease accompanying MCs [1,9,10], but other injury patterns associated with MCs include autoimmune disease, low virulence bacterial infections, and microtrauma resulting in endplate abnormalities [11,12,13].

Given that MCs are associated with advanced spondylosis, studies have attempted to associate MCs with axial neck and back pain [14,15,16]. Some authors have even attempted to correlate type I MC with accelerated degeneration of the adjacent intervertebral disc [17]. However, while the severity of type I MCs have been linked to worse pain symptoms in observational studies, systematic reviews have not consistently substantiated this finding [18]. A meta-analysis in 2016 attempted to identify trends in outcomes based on pre-operative MCs and their effect on surgical outcomes. While they found a trend toward worse improvement in lumbar discectomy patients who had MCs, they concluded that it was unlikely that the difference met the minimal clinically important difference (MCID) threshold [19]. However, no previous study has performed a meta-analysis on patients undergoing cervical spine surgery, and additional literature has been published on patients with MCs undergoing lumbar discectomy since 2016 [20,21,22,23]. Therefore, our objective was to evaluate the impact of MCs on neck and back pain in patients undergoing cervical or lumbar spine surgery.

This research received no external funding. All data are contained within the article. The authors declare no conflict of interest.

## 2. Materials and Methods

This systematic review follows the Preferred Reporting Items for Systematic Reviews and Meta-Analyses (PRISMA) Statement (Table A1). This study was IRB exempt since only published studies were incorporated. A systematic literature review was performed of the PubMed database from its inception until 1 May 2022. “Modic changes” or “endplate signal changes” coupled with “outcomes” were queried. To ensure the inclusion of all of the available evidence, the references of each study meeting the inclusion criteria were searched to identify additional studies that merited inclusion.

### 2.1. Study Eligibility

Studies were included if MCs, identified in preoperative MRI, were operatively treated in either the cervical or lumbar spine. Studies were excluded if (1) clinical follow-up time was not reported or it was less than one year; (2) a full text manuscript could not be obtained; (3) the article was not written in or translated to English; or (4) the article was a letter to editor or systematic review. 

Two reviewers independently screened the identified articles and selected studies for full-text review after screening of the title and abstract. Articles were screened for inclusion based on the predetermined inclusion/exclusion criteria during the full-text review. A search of the references was performed of all articles meeting the inclusion criteria to identify any potentially missed manuscripts. For studies where article inclusion was unclear, a more senior author was consulted to resolve any discrepancies. 

### 2.2. Data Collection Quality Assessment

The authors extracted all potentially relevant data from the identified studies with multiple self-designed tables. Data including the type and prevalence of Modic signal changes, clinical outcome measures, patient population, procedure type, and key study findings were reported based on the affected region of the spine. Studies were assessed for bias using the validated Newcastle–Ottawa Score with a high-quality score defined as a score greater than or equal to 6 [24].

### 2.3. Meta-Analysis

Due to significant data heterogeneity, variability in the clinical outcome measures assessed, and the different surgical procedures performed in the lumbar spine, only a limited meta-analysis could be performed. For meta-analysis inclusion, a minimum of four studies were required to report on the same patient reported outcome (PRO) with only four PROs meeting this criteria (Visual Analog Scale (VAS) Back, VAS Neck, Oswestry Disability Index (ODI), and Japanese Orthopedic Association (JOA) score). The weighted mean difference in values between patients with and without MCs was then assessed using forest plots and data heterogeneity was calculated as an *I*^2^ value. The meta-analysis data were generated using R Studio Version 4.0.2 (Boston, MA, USA).

## 3. Results

The initial PubMed search identified 259 articles, after which the title and abstract screening identified 36 potentially relevant articles. Two were excluded due to an inadequate follow-up of less than one year [25,26], another two were excluded because of the risk of potentially identical patients [27,28], and one article’s full-text could not be obtained [29]. Thirty-one articles were identified after a full-text review. Two studies were classified as a high risk of bias, so they were also excluded [30,31]. In total, 29 articles were included in the final analysis (Figure 1). The characteristics of the studies included in the systematic review are presented in Table 1 and Table 2. The included articles comprised a total of 1871 patients with cervical spine disease, 683 (36.5%) of whom displayed some type of Modic change and 4142 patients with lumbar disease, 2005 (48.4%) of whom displayed some type of Modic change.

**Figure 1 ijerph-19-10158-f001:**
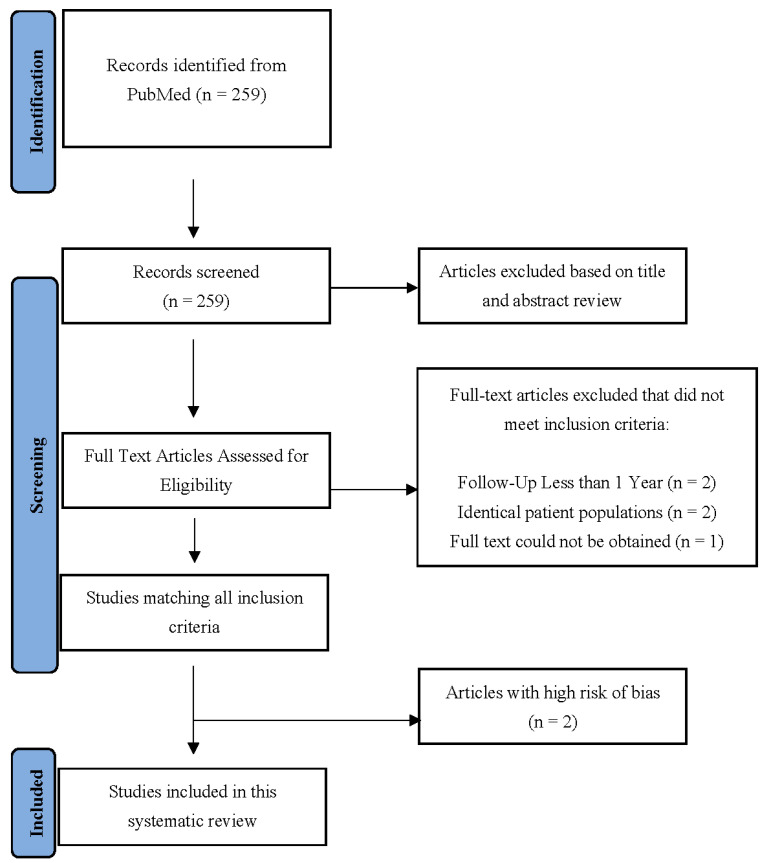
The PRISMA flow diagram describing article selection for inclusion.

**Table 1 ijerph-19-10158-t001:** The patient characteristics of studies evaluating the cervical spine.

Authors	Patient Population	Modic Subtypes	Age	Gender	Patients	Any MC (%)	MC-I (%)	MC-II (%)	Risk of Bias
Yang et al. [32] (2019)	Patients with cervical radiculopathy due to single-level disc herniation	I, II	45.2 (7.3)	131	223	41 (18.4%)	10 (4.5%)	29 (13%)	7
Baker et al. [33] (2020)	Patients with symptomatic degenerative pathology refractory to conservative management	I, II, III	NR	NR	861	356 (41.3%)	70 (8.1%)	218 (25.3%)	9
Huang et al. [34] (2020)	Patients who underwent single-level ACDF with MC-II	II	50.4 (1.6)	58	116	24 (20.7%)	0	24 (20.7%)	8
Li et al. [35] (2015)	Patients who underwent single-level ACDF with MC-II	II	47.0 (7.2)	134	248	35 (14.1%)	0	35 (14.1%)	7
Li et al. [36] (2017)	Patients with chronic axial symptoms resulting from single-level radiculopathy or myelopathy	II	56.1 (6.1)	36	76	76 (100%)	0	76 (100%)	7
Zhou et al. [37] (2018)	Patients with cervical spondylotic myelopathy	NR	56.1 (7.3)	56	117	28 (23.9%)	NR	NR	8
Li et al. [38] (2022)	Patients with MCs cervical spondylotic myelopathy with hernia behind the vertebrae or OPLL	I, II	55.0 (22.2)	67	124	61 (49.2%)	20 (16.1%)	41 (33.1%)	6
Li et al. [39] (2015)	Patients with chronic axial symptoms resulting from single-level cervical disk degeneration nonresponsive to appropriate nonsurgical treatment for at least 6 months	I, II	55.8 (6.5)	49	106	62 (58.5%)	23 (21.7%)	39 (36.8%)	7

MC—Modic changes; MC-I—type I Modic change; MC-II—type II Modic change; ACDF—anterior cervical discectomy and fusion; OPLL—ossification of the posterior longitudinal ligament; NR—not reported.

**Table 2 ijerph-19-10158-t002:** The patient characteristics of studies evaluating the lumbar spine.

Authors	Patient Population	MC Subtypes	Age	Gender	Patients	Any MC (%)	MC-I (%)	MC-II (%)	Risk of Bias
Kumarasamy et al. [21] (2021)	Patients with LBP and single level lumbar disc herniation	I, II	42.5 (12.6)	107	309	86 (27.8%)	6 (1.9%)	68 (22%)	7
Jiao et al. [40] (2021)	Patients with LBP and either LDH, spinal stenosis, or spondylolisthesis who underwent single-segment TLIF with a PEEK cage	I, II	56.7 (8.9)	49	89	51 (57.3%)	20 (22.5%)	31 (60.8%)	6
el Barzouhi et al. [41] (2014)	Patients with sciatica	I, II	43.2 (10.1)	140	263	112 (42.6%)	4 (1.5%)	106 (40.3%)	8
Ulrich et al. [42] (2020)	Patients with claudication and lumbar stenosis	I, II	66.8 (6.3)	96	205	143 (69.8%)	22 (15.4%)	93 (65.0%)	8
Chung et al. [43] (2021)	Patients with lumbar DDD	I, II, III	64.7 (9.1)	54	86	NR	NR	NR	7
MacLean et al. [23] (2021)	Patients with single level LDH	I, II, III	53 (13)	101	179	110 (61.5%)	28 (15.6%)	63 (35.2%)	7
Udby et al. [44] (2020)	Patients with bilateral or unilateral radiculopathy	I, II	50.5	310	620	290 (46.8%)	73 (11.8%)	217 (35%)	7
Sørlie et al. [45] (2012)	Patients with one-level lumbar disc herniation	I, II	41.2 (12.1)	66	178	104 (58.4%)	36 (20.2%)	68 (38.2%)	8
Gornet et al. [46] (2014)	Patients with back pain due to DDD with pre-op ODI ≥ 30	I, II	NR	NR	89	NR	NR	NR	8
Ohtori et al. [47] (2010)	Patients with LBP and leg pain due to lumbar spinal canal stenosis	I, II	65.4	16	33	33 (100%)	21 (63.6%)	12 (36.4%)	6
Cao at al [48] (2014)	Patients with one-level LDH and MCs	I, II	NR	NR	91	91 (100%)	42 (46.2%)	60 (65.9%)	7
Lurie et al. [49] (2013)	Patients with radicular pain due to intervertebral disc herniation	I, II	41.7 (11.4)	522	307	80 (26.1%)	27 (8.8%)	53 (17.3%)	7
Xu et al. [50] (2019)	Patients with unilateral radicular pain due to one-level intracanal disc herniation	I, II	40.0 (12.5)	104	276	94 (34.1%)	44 (15.9%)	50 (18.1%)	6
Djurasovic et al. [51] (2012)	Patients with “disc pathology” listed as primary surgical indication	I, II, III	47	23	51	NR	NR	NR	7
Masala et al. [52] (2014)	Patients with LBP without radicular symptoms unresponsive to conservative therapy for 6 months with type I MC	I	40.3 (8.2)	133	218	218 (100%)	218 (100%)	0	6
Ohtori et al. [53] (2010)	Patients with one-level LDH	I	35.5	19	45	23 (51.1%)	23 (51.1%)	0	6
Rahme et al. [54] (2010)	Patients with one-level LDH	I, II, III	54	14	41	32 (78%)	6 (14.6%)	26 (63.4%)	7
Blondel et al. [55] (2011)	Patients with chronic LBP due to single-level DDD	I, II	42.1	101	221	114 (51.6%)	65 (29.4%)	49 (22.2%)	8
Gautschi et al. [56] (2016)	Patients with LBP due to disc herniation, spinal stenosis or DDD requiring lumbar fusion	I, II, III	58.6 (15.5)	144	338	202 (59.8%)	NR	NR	7
Hellum et al. [57] (2012)	Patients with LBP due to LDD with an ODI ≥ 30%	I, II	41.2 (7.0)	81	152	131 (85%)	48 (31.6%)	55 (36.2%)	9
Kwon et al. [58] (2009)	Patients who underwent PLIF	I, II, III	47.4	232	351	92 (26.2%)	26 (7.4%)	55 (15.7%)	7

MC—Modic changes; MC-I—type I Modic changes; MC-II—type II Modic change; LBP—low back pain; LDH—lumbar disc herniation; TLIF—transforaminal lumber interbody fusion; PEEK—polyetheretherketone; DDD—degenerative disc disease; ODI—Oswestry Disability Index; LDD—lumbar disc disease; PLIF—posterior lumbar interbody fusion; NR—not reported.

### 3.1. Cervical Spine

Eight retrospective studies described the correlations between MC and surgical outcomes (Table 3). No prospective studies evaluated MC in the cervical spine. In general, studies did not report a difference in the PROs based on the presence of MC. 

Huang et al. [34] reviewed 116 cases of single-level anterior cervical discectomy and fusions (ACDF) for the presence of MC and their association with fusion rates and PROs. Patients with type II MC experienced significantly delayed early fusion rates at 3- and 6-months, but the fusion rates were similar at one year. There were no significant differences between groups with regard to improvement in the VAS scores or JOA scores.

In a study by Li et al. [38] in 2022, 124 patients underwent single-level anterior cervical corpectomy and fusion for cervical myelopathy. They found that 41% of patients with MC had cage subsidence, defined as at least 1 mm, compared to only 15.9% of patients without MC (*p* = 0.003). Subsidence did not vary between the Modic subtypes. Patients with type I MC had a higher proportion of partial fusions, defined by incomplete bony remodeling (40% compared to 11% of controls). No cases of pseudarthrosis were observed and no other differences were identified with regard to the JOA, neck disability index (NDI), and VAS neck and arm pain scores.

Yang et al. [32] retrospectively reviewed patients who underwent one-level ACDFs or cervical disc arthroplasty. The presence of MCs did not result in different one-year VAS neck, NDI, or physical component summary (PCS-12) and mental component summary (MCS-12) scores from the 36-Item Short Form Health Survey (SF-36). 

In a 2017 study by Li et al. [36], 72 patients were analyzed with type II MC and minimum five-year follow-up. Of those, 35 received cervical disc arthroplasty and 37 underwent ACDF for myelopathy or radiculopathy. All patients experienced postoperative improvement assessed by modified JOA (mJOA), NDI, and VAS, without differences between groups. The patients in the disc replacement group were noted to have improved range of motion at final follow-up.

In 2015, Li et al. [35] retrospectively compared 35 patients who had a single-level ACDF with type II MC at an adjacent level to 213 patients without type II MC. They observed no significant differences between groups with regard to the range of motion or disc height at adjacent levels, and no differences in improvement by the mJOA and NDI scores. The patients in the MC group had worse VAS neck pain at one-year follow-up (*p* < 0.05), although these differences resolved by 5-year follow-up.

A retrospective review of 117 patients with a single-level ACDF found that preoperative MCs at levels adjacent to surgery were associated with greater axial symptoms (*p* = 0.015) [37]. However, all patients demonstrated clinical improvement as assessed by the JOA without a significant difference between patients who did and did not have MCs. 

In a separate 2015 study by Li et al. [39], 106 patients were retrospectively identified with one-level ACDFs for cervical spondylotic myelopathy. Preoperatively, 23 patients had type I MC, 28 had type II MC, and 44 had no MC. The patients in the type I MC group had significantly lower VAS neck pain at 24 months after surgery (1.5 ± 1.1) compared to patients without (MC 2.0 ± 1.5), although a specific *p*-value was not provided. 

Baker et al. [33] retrospectively examined the records of 861 patients who underwent a one- to four-level ACDF for radiculopathy or myelopathy. Of those, 365 patients had MCs. No significant differences in postoperative NDI, VAS-neck, or VAS-arm were found between groups. However, after stratification by cervical level, they identified the presence of MC at C7–T1 predicted worse postoperative NDI (*p* < 0.001). 

A limited number of studies collected PROs in a standardized manner, thus, a meta-analysis could only be performed for postoperative values. Four studies reported on neck pain and JOA following ACDF [34,35,38,39]. Random effects analysis of these studies demonstrated no significant association between MC and postoperative VAS neck scores (MD −0.17, 95% CI: −0.50–1.70) (Figure 2a) or the JOA scores (MD −0.07, 95% CI: −0.35–0.20) (Figure 2b). Among these studies, moderate heterogeneity in neck pain scores (I^2^ = 52%) and low heterogeneity in JOA scores (I^2^ = 0%) were observed.

### 3.2. Lumbar—Discectomy/Microdiscectomy

There were a total of 11 articles discussing the outcomes after discectomy or microdiscectomy with three articles comparing those procedures to lumbar fusion (Table 4). Regardless of the presence of MC, patients experienced significant improvement following surgery. However, the studies delivered mixed results as to whether MCs modified the magnitude of benefit obtained from surgery. 

El Barzouhi et al. [41] conducted a multicenter, randomized trial of 283 patients with sciatica. They compared microdiscectomy to conservative care. Of the 283 patients, 41 percent of patients had vertebral endplate signal changes (VESCs). No significant differences in the amount of recovery or disabling back pain were identified between patients with and without VESCs at one year follow-up.

Gautschi et al. [56] prospectively identified 338 patients with lumbar degenerative disc disease. Of the 338 patients, 175 of them had a microdiscectomy. Similar outcomes for VAS back and leg pain, ODI, Roland–Morris Disability Index, Timed Up and Go test, EuroQol-5D (EQ-5D), and PCS-12 were identified between patients in the MC and no-MC cohorts.

Kumarasamy et al. [21] prospectively followed 309 patients undergoing microdiscectomy, 86 of whom had MC. The patients in the MC group had worse postoperative numeric rating scale (NRS) back pain (1.6 to 1.1, *p* = 0.001), although the overall improvement in both groups was 4.3 points, indicating that there was no difference in the magnitude of overall improvement. Both groups experienced a significant improvement in disability, but the group without MC experienced statistically significant greater improvement, although this did not meet the MCID threshold. Patient satisfaction, as evaluated by the MacNab criteria, was noted to be significantly worse for patients with MCs.

Sørlie et al. [45] prospectively evaluated 178 patients undergoing microdiscectomy and found that 36 patients had type I MCs. Multivariate analysis demonstrated no statistical differences between groups for VAS back and leg pain, ODI, and EQ-5D. All patients had significant improvement after surgery, although patients who smoked and had type I MCs had less improvement. 

Rahme et al. [54] retrospectively identified patients treated with lumbar microdiscectomy. They included 41 patients in the final analysis and identified 19 patients with preoperative MC at the level of the operation. At median follow-up of 41 months, 32 patients had MCs (since four of the 22 patients without MCs converted to type I MCs and another nine converted to type II MCs). Additionally, 60% of the type I MCs converted to type II MCs. The study concluded that the majority of patients after a lumbar discectomy will convert to type II MCs, but MCs do not results in worse outcomes, recurrent symptoms, or decreased satisfaction.

Ohtori et al. [53] prospectively identified 23 patients with type I MC undergoing lumbar discectomy and matched them to 22 patients without MC. All patients exhibited improvement in VAS back, JOA, and ODI with no significant differences between the groups at 12 or 24 months after surgery.

MacLean et al. [23] retrospectively analyzed 129 patients undergoing discectomy for lumbar radiculopathy with 77 patients having MCs. They had complete outcome data for 96 of these patients and concluded each group experienced statistically and clinically significant improvement as assessed by the PCS-12, ODI, and VAS leg pain scores. Most subgroups met the MCID and there were no significant differences in clinical improvement based on the presence or absence of MCs. 

Lurie et al. [49] retrospectively reviewed 307 patients treated with discectomy for radicular pain and identified 81 patients with MC. The patients with type I MC improved significantly less in ODI compared to the type II or no MC patients. However, patients with type I MC did experience an average improvement in ODI of 26.4 points following discectomy.

Udby et al. [44] retrospectively reported on 620 patients undergoing lumbar discectomy for radiculopathy with 290 patients having MC. At two year follow-up, they concluded that both groups had significant improvement by ODI, EQ-5D, VAS back and leg pain, with no significant differences between groups.

Cao et al. [48] retrospectively reviewed 91 patients with lumbar disc herniation and MC who had a combination of low back and radicular leg pain. A total of 47 patients were treated with discectomy, while 44 underwent a posterior lumbar interbody fusion (PLIF). All patients had significant VAS back pain improvement; however, the patients in the PLIF group experienced greater VAS back improvement. Both groups had significant VAS leg pain improvement and there was no postoperative difference based on the type of surgery. 

Xu et al. [50] retrospectively reviewed 276 patients undergoing percutaneous endoscopic lumbar discectomy for radiculopathy and identified 94 patients with MC. They found no differences in VAS leg pain through final follow-up of 30 months. At three months follow-up, all patients had similar improvement in ODI and VAS back pain, however, the patients with MCs were found to have worse trending ODI and VAS back pain at final follow-up. 

Ulrich et al. [42] retrospectively identified 205 patients undergoing lumbar discectomy or lumbar fusion. They found no differences in the outcomes between patients with MCs versus those without MCs, regardless of the procedure type when assessing the Spinal Stenosis Measure, EQ-5D, and NRS back pain scores at the 36 month post-surgical visit. Over 70% of patients reached the postoperative MCID at 36 month follow-up. Thus, the study concluded that MCs do not significantly affect the postoperative clinical outcomes.

Five studies reported values for ODI [21,23,44,45,50], and four studies reported VAS back pain [21,44,45,50]. A meta-analysis was performed among these studies and identified no significant associations between MC and either postoperative VAS back pain or ODI (Figure 3). However, significant data heterogeneity was present for VAS and ODI (I^2^ = 88% and I^2^ = 96%, respectively), indicating that additional high quality studies are required to confirm the finding that MCs do not affect the improvement in VAS back or ODI scores postoperatively.

### 3.3. Lumbar Fusion

Nine studies evaluating lumbar fusion surgery were identified. One study evaluated oblique lateral interbody fusions (OLIF) [43], two evaluated transforaminal lumbar interbody fusions (TLIFs) [23,40], two included multiple fusion techniques [51,56], two evaluated PLIF [42,47], and two evaluated posterolateral fusions [48,58]. In general, patients had substantial improvements in the clinical outcomes and fusion rates regardless of the presence of MCs, with the exception of one study, which found decreased fusion rates and worse outcomes in patients with type III MCs [58]. 

Chung et al. [43] retrospectively analyzed 86 patients with 125 operated levels by one- or two-level OLIF and identified MC at 72 of these levels. They found no association between the MC and fusion rate or cage subsidence at 28 months after surgery.

Jiao et al. [40] retrospectively identified 89 patients, 51 with MC, who underwent single-level TLIFs. The authors found no difference in the fusion rate or clinical outcomes based on MCs, but type I MCs were associated with significantly higher rates of cage subsidence (40% to 15.8%). All other outcome measures were similar between groups including disc height, segmental lordosis, and lumbar lordosis. 

MacLean et al. [23] retrospectively reviewed 44 patients who underwent TLIF for radiculopathy and instability. They concluded that patients experienced significant improvement by PCS-12, ODI, and VAS leg pain regardless of the presence of MCs.

Two studies reported on multiple fusion techniques: TLIF, PLIF, PLF, extreme lateral interbody fusion, circumferential fusion, and anterior lumbar interbody fusion [51,56]. Between these studies, 121-disc levels were included in their analysis with 73 levels having MCs. No outcome measures were significantly affected by MCs including ODI, VAS back and leg, or SF-36.

Of the two studies reporting on decompression with PLIF, Ohtori et al. [47] prospectively evaluated 33 patients with MC while Ulrich et al. [42] retrospectively identified 57 patients undergoing lumbar fusion with 41 having MCs and 16 having no MCs. All patients in the study by Ohtori et al. [49] had solid arthrodesis at 24 months after surgery with average union at nine months. Neither study found differences in the VAS nor ODI based on the presence of MCs.

For patients undergoing posterolateral fusion, Kwon et al. [58] retrospectively reviewed 351 patients and identified MCs in 92 while Cao et al. [48] retrospectively identified 44 patients with MCs. Kwon et al. reported lower fusion rates for patients with type III MCs (54.5%) compared to those without MCs (96.5%) at a minimum three years follow-up. They also identified less improvement for those with type III MC when evaluated by Prolo’s scale and VAS scores. Cao et al. reported that all patients had significant improvement in the VAS back, VAS leg, and JOA. 

### 3.4. Other Lumbar Surgeries

We identified four articles in our search that did not fit into the above categories. Three of these studies reported on disc arthroplasty while the other study reported on vertebroplasty. Masala et al. [52] prospectively identified 218 patients with type I MC who underwent vertebroplasty. Of those, 98% showed improvement in VAS and ODI, 1% showed no improvement, and 1% died for unrelated reasons. Three prospective studies evaluated lumbar disc arthroplasty [46,55,57]. They all reported improved ODI in patients with MC compared to those without. Gornet et al. [46] reported these findings for type II MC, Blondel et al. [55] reported these findings for type I MC, and Hellum et al. [57] reported these findings for both type I and II MCs.

## 4. Discussion

Radiographic evidence of intervertebral disc degeneration is reported to be present in greater than 90% of individuals by the time they reach 50 years of age [59]. Most patients with disc degeneration remain asymptomatic, but within the asymptomatic population, 21% of patients older than 60 have spinal stenosis, while 36% have a herniated intervertebral disc [60]. Therefore, identifying potential factors that may result in back or neck pain is prudent [61]. Modic changes are one potential etiology for symptomatic back and neck pain, thus an evaluation of their potential improvement after surgery is indicated. Although the heterogeneity in the literature precluded our team from performing a meta-analysis of all of the included studies, the majority of studies assessing the surgical outcomes after cervical and lumbar spine surgery have not detected clinically important differences between patients with and without Modic changes. Despite heterogeneity, our meta-analysis similarly identified no differences between postoperative VAS back pain or ODI in patients with MC undergoing lumbar discectomy. There was less data heterogeneity in patients undergoing ACDF, and our meta-analysis found no differences in postoperative VAS neck pain or JOA based on the presence of MCs.

Surgeons perform over 100,000 ACDFs each year [62] and MCs are present at rates surpassing 40% in patients greater than 50 years of age who undergo cervical spine MRI [63]. The results of our pooled analysis found that MCs were present in a similar 36.5% of patients who required cervical spine surgery. Thus, understanding the potential indicators of surgical success (e.g., presence or absence of MCs) is important for risk stratification and guiding the patient’s expectations on their potential postoperative clinical improvement. When evaluating the clinical outcomes, our pooled data suggest that MCs are not predictive of postoperative improvement in patients undergoing cervical spine surgery. While one study reported MCs at the level C7–T1 were associated with worse outcomes, this group comprised 1.2% of their overall patient population and was likely severely underpowered to detect a true difference [33]. In contrast, Li et al. [39] reported that patients with type I MC were more likely to have reduced back pain at 2.5 year follow-up. Given that the available data are based on retrospective studies, with approximately half the patients coming from one study [33], it is likely that additional well-designed studies are needed to definitively claim that MCs have no significant clinical effect on cervical spine postsurgical outcomes. However, the evaluated studies do pose a low risk of bias, and the current literature suggests that MCs do not significantly impact the clinical or surgical success of cervical spine surgery [34,38]. 

Seven studies evaluated the microdiscectomy or discectomy outcomes in relation to MC. While discectomy and microdiscectomy overall appear efficacious as a treatment for axial back pain reduction and PRO improvement for patients with MCs, those with type I MC may experience reduced improvements following discectomy. Only one study evaluating microdiscectomy identified worse improvement in patients with MCs, although this was not clinically significant [21]. Two other retrospective studies identified a relationship between MC and worse PRO improvement (ODI and VAS back) following discectomy. Although these studies identified discectomy as an effective treatment option for patients with MC, patients with type I MC experienced less improvement at long-term follow-up [49,50]. Type I MC likely represents inflammatory changes, which may be linked to exacerbations of low back pain. Despite the propensity of MC to transform to other types, they have also been shown to remain constant over time [64]. Although discectomy may alleviate discogenic pain, it may have little effect on the concomitant changes to the vertebral endplate. This may be one potential reason why patients with MC-I do not exhibit as robust of an improvement when compared to other patients. While the results of our meta-analysis indicated no differences in back pain or disability, significant heterogeneity and the limited number of studies prohibit strong conclusions, and more research is needed, especially comparing the improvement across MC types.

We identified eight retrospective studies and one prospective study reporting MC in lumbar fusion surgery. These studies demonstrated that MC do not influence patient-reported outcomes or fusion rates following lumbar fusion surgery regardless of approach. Jiao et al. [40] identified that patients with type I MC experienced greater improvement in PROs following TLIF. Kwon et al. [58] reported lower fusion rates and worse PROs in type III MC. They suggested that given the sclerotic nature of type III MC, additional fusion such as pedicle screw fixation may be required for patients with type III changes [58]. Although this sclerotic bone may interfere with fusion, resorption of the sclerotic bone has been observed over time [65]. Care should be taken when selecting patients for surgery with MC-III because this population remains understudied due to its low prevalence. While lumbar disc replacement was evaluated in three prospective analyses and all three demonstrated excellent results that are not affected by the presence of MC, further research is needed to analyze the impacts of disc arthroplasty on endplate disease.

## 5. Conclusions

Since the last review on this topic in 2016, several prospective and large retrospective studies have been published with a low risk of bias [19]. After summarizing all of the available data on surgical patients with MCs, quantitative and qualitative analyses suggest that patients with MCs have similar clinical improvements in PROs (ODI and VAS back/neck) and similar fusion rates when compared to patients without MCs. However, additional high-quality studies are needed to further elucidate changes in the fusion status, especially among patients with type III MC. Furthermore, long-term prospective studies evaluating the outcomes of patients with type I MC undergoing discectomy are merited given that some studies indicate that they have worse clinical improvement after surgery.

## Figures and Tables

**Figure 2 ijerph-19-10158-f002:**
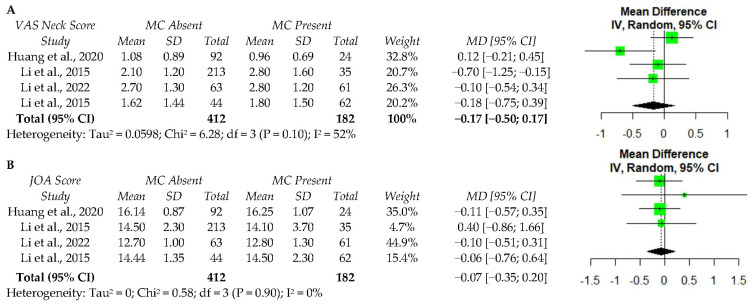
The meta-analysis of one-year postoperative VAS neck score (**A**) and JOA score (**B**) following ACDF. VAS—visual analog scale; MC—Modic changes; SD—standard deviation; MD—Mean difference; CI—confidence interval; JOA—Japanese Orthopaedic Association score; df—degrees of freedom; IV—independent variable [34,35,38,39].

**Figure 3 ijerph-19-10158-f003:**
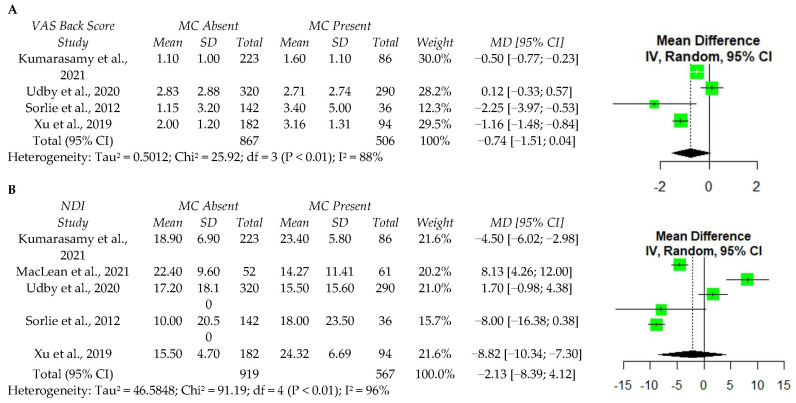
A meta-analysis of one year postoperative VAS back score (**A**) and ODI (**B**) following discectomy. VAS—visual analog scale; MC—Modic changes; SD—standard deviation; MD—Mean difference; CI—confidence interval; NDI—Neck Disability Index; df—degrees of freedom; IV—independent variable [21,23,44,45,50].

**Table 3 ijerph-19-10158-t003:** The key findings of surgery in patients with cervical Modic changes.

Authors	Objective	Study Design	Procedure	Follow-Up (Months)	Clinical Outcome Measures	Key Findings
Yang et al. [32] (2019)	To report on the incidence of MC in patients with cervical radiculopathy due to disc herniation	Retrospective	ACDF vs. ACDA vs. ACD	12	NDI, MCS-12, PCS-12, Neck VAS, Arm VAS	MCs were not associated with a change in NDI, SF-12, VAS Surgical approach did not influence MRI-evidence of MCs
Baker et al. [33] (2020)	To study the association of MC with postoperative outcomes in ACDF patients	Retrospective	ACDF	27.3	VAS Neck, Vas arm, SF12, VR12	Overall, MCs were not associated with post-operative PROs
Huang et al. [34] (2020)	To explore the impact of MC on bone fusion after single-level ACDF	Retrospective	ACDF	33.2	JOA score, VAS neck, fusion rates	MCs were not associated with post-operative PROs, but MC-II were associated with delayed fusion.
Li et al. [35] (2015)	To explore the impact of MC-II on the clinical outcomes of single-level ACDF	Retrospective	ACDF	60	JOA, NDI, neck VAS	MCs were not associated with post-operative PROs or fusion rates
Li et al. [36] (2017)	To compare the clinical and radiologic outcomes of patients with MC-II who underwent single level ACDF or ACDA	Retrospective	ACDF vs. ACDA	60	JOA, NDI, ROM, VAS neck, VAS arm	All patients improved from baseline, but the ACDA group showed greater improvement in VAS neck and axial ROM compared with the ACDF group at final follow-up.
Zhou et al. [37] (2018)	To compare the clinical and radiological outcomes between patients with or without axial symptoms in ACDF	Retrospective	ACDF	12	Axial symptoms	Patients with post-operative axial symptoms were more likely have had preoperative MCs on endplates adjacent to treated disc
Li et al. [38] (2022)	To determine the impact of MCs on cage subsidence and fusion after ACCF	Retrospective	ACCF	24	Cage subsidence, fusion rate	More patients with MCs experienced cage subsidence. MCs were not associated with post-operative PROs or fusion rates.
Li et al. [39] (2015)	To analyze the influence of MCs on the clinical results of cervical spondylotic myelopathy treated by ACDF	Retrospective	ACDF	24	JOA, percent recovered at final follow-up visit	All patients experienced significant improvement in all measures. MC-I patients reported significantly lower VAS at 3, 6, 12, and 24 months postop. MC-I patients had a higher JOA at 1-year

MC—Modic changes; MC-I—type I Modic change, MC-II—type II Modic change; ACDF—anterior cervical discectomy and fusion; ACDA—anterior cervical discectomy with arthroplasty; ACD—anterior cervical discectomy without intervertebral cage; NDI—neck disability index; MCS-12—mental component score from the 12-item short form survey; PCS-12—physical component score from the 12-item short form survey; VAS—visual analog scale; SF-12—12-itme short form survey; PROs—patient reported outcomes; JOA—Japanese Orthopedic Association; ROM—range of motion; ACCF—anterior cervical corpectomy and fusion.

**Table 4 ijerph-19-10158-t004:** The key findings of surgery in patients with lumbar Modic changes.

Authors	Objective	Study Design	Procedure	Follow-Up (Months)	Outcome Measures	Key Findings
Kumarasamy et al. [21] (2021)	To evaluate the relationship between MC and clinical outcomes after a lumbar microdiscectomy	Prospective	Microdiscectomy	12	NRS pain, ODI, patient satisfaction by Mac Nab criteria	Patients with MC had less improvement in back pain and ODI scores at all follow-ups. However, MCID between the groups was not significant
Jiao et al. [40] (2021)	To analyze the influence of MCs the clinical and radiographic outcomes of transforaminal lumbar interbody fusion	Retrospective	TLIF	23.4	ODI, VAS back pain, VAS leg pain, cage subsidence,	MCs had no impact on fusion rates and clinical outcomes
el Barzouhi et al. [41] (2014)	To analyze the correlation between MCs and back pain in sciatica in patients with early surgery vs. conservative treatment	Prospective	Microdiscectomy	12	VAS back, 7-point Likert self-rating scale of global perceived recovery	Surgically treated patients showed an increase in extent of MCs (67% of pts) compared to conservatively treated patients (19%) No difference in post-operative back pain scores
Ulrich et al. [42] (2020)	To investigate if the MCs are predictive for outcomes in degenerative lumbar spinal stenosis patients undergoing decompression-alone or decompression with instrumented fusion surgery	Retrospective	Decompression vs. PLIF	36	SSM symptoms, SSM function, MCID in SSM symptoms, NRS pain, and EQ-5D sum score over time	MCs were not associated with clinical outcomes, independent of the chosen surgical operation.
Chung et al. [43] (2021)	To evaluate the influence of MC on the radiological outcomes in lumbar interbody fusion	Retrospective	OLIF	28.6	Cage subsidence, fusion rate	MCs were not associated with cage subsidence or impaired fusion
MacLean et al. [23] (2021)	To examine the relationship between preoperative MCs and postoperative clinical assessment scores for patients receiving lumbar discectomy or TLIF for lumbar disk herniation	Retrospective	TLIF vs. discectomy	12	VAS leg, SF12 physical, ODI	All patients experienced improved from baseline, but those with MC experienced the greatest improvement in disability. Outcomes were similar in discectomy vs. TLIF
Udby et al. [44] (2020)	To assess whether MCs are associated with health-related quality of life, long-term physical disability, back- or leg pain after discectomy	Retrospective	Discectomy	24	ODI, VAS back, VAS leg, Patient satisfaction scores, EQ-5D	MCs were not associated with differences in improvement in PROs, except for VAS back wherein patients with MC-I had worse scores than those with MC-II
Sørlie et al. [45] (2012)	To investigate whether the presence of preoperative MC-I represents a risk factor for persistent back pain 12 months after surgery amongst patients operated for lumbar disc herniation	Retrospective	Microdiscectomy	12	VAS back, VAS leg, ODI, EQ-5D, self-reported benefit of the operation and employment status	All patients improved in all outcomes at 1-year. In aggregate, MC were not associated with PROs. Patients with MC-I had less improvement of VAS Back and EQ-5D
Gornet et al. [46] (2014)	To determine which variables predict clinical outcomes following disc replacement	Prospective	Disc replacement	60	ODI, SF-36	Patients with MC-II had better ODI scores at 5-year follow-up than those with no-MC or MC 1
Ohtori et al. [47] (2010)	To investigate the changes in MCs after posterolateral fusion	Prospective	Posterolateral fusion	24	JOA, VAS back, ODI, fusion rate	MCs were not associated with post-operative PROs or fusion rates
Cao at al [48] (2014)	To compare the outcomes of simple discectomy and instrumented PLIF in patients with lumbar disc herniation and MCs	Retrospective	Instrumented PLIF vs. discectomy	18	JOA, VAS back, VAS leg	iPLIF resulted in superior outcomes for relief of LBP compared to simple discectomy. Both treatments similarly relieved radicular leg pain
Lurie et al. [49] (2013)	To determine whether baseline MRI and MCs are associated with differential outcomes with surgery or non-operative treatment	Retrospective	Open discectomy and decompression	48	ODI, bodily pain, sciatica and back pain symptoms, physical function	MC-I patients had poorer outcomes on all measures after surgery compared to MC-II or no MC
Xu et al. [50] (2019)	To assess the clinical outcomes of TF-PELD in the treatment of LDH and MCs	Retrospective	TF-PELD	29.6	ODI, VAS back, VAS leg, Patient satisfaction scores (Modified MacNab)	Patients with MC-I had poorer improvement in VAS back and ODI at 1 year and final follow-up compared to MC-II or no MC. Improvements in leg pain were comparable among groups
Djurasovic et al. [51] (2012)	To investigate relationship between MRI findings in patients with DDD and clinical improvement after lumbar fusion	Retrospective	PLF, TLIF, ALIF, circumferential fusion	24	NRS back and leg, ODI, SF-36	MCs were not associated with post-operative PROs
Masala et al. [52] (2014)	To evaluate the effectiveness of vertebral augmentation with calcium sulfate and hydroxyapatite resorbable cement in patients with LBP due to MC-I	Prospective	Vertebroplasty with calcium sulfate and hydroxyapatite resorbable bone cement	12	VAS back, ODI	All patients experienced improvement in pain and disability
Ohtori et al. [53] (2010)	To examine the relationship between LBP after discectomy for disc herniation and MC 1	Prospective	Discectomy	24	VAS back, ODI, JOA	All scores improved from baseline. MC were not associated with post-operative PROs or fusion rates
Rahme et al. [54] (2010)	To study the impact of surgery on the natural history of MC	Retrospective	Discectomy	60	ODI, patient satisfaction, presence of symptoms, work status	MC were not associated with post-operative clinical outcomes
Blondel et al. [55] (2011)	To analyze the influence of MC on the clinical results of lumbar total disc arthroplasty	Prospective	Disc replacement	30	VAS back, VAS leg, ODI	All groups improved in all outcomes at final follow-up. Patients with MC1 had the greatest improvement in ODI and radicular pain by final follow-up
Gautschi et al. [56] (2016)	To determine the relationship of radiological grading scales of lumbar DDD with postoperative pain intensity, functional impairment, and health-related quality of life	Prospective	Microdiscecomty, decompression, TLIF, PLIF, or XLIF	24	ODI, RMDI, SF-12, PCS-12, and EQ-5D index	No significant difference in improvement in clinical outcome between patients with or without MC
Hellum et al. [57] (2012)	To evaluate predictors of outcome in patients treated with disc prosthesis or multidisciplinary rehabilitation	Prospective	Disc replacement	24	ODI	Patients with MC-I or MC-II has significantly better ODI outcomes after disc replacement
Kwon et al. [58] (2009)	To investigate the efficacy of PLIF with cages in chronic DDD with MCs	Retrospective	PLIF w/cage	59.8	Fusion rate, Prolo’s scale for symptomatic improvement, VAS Back	Patients with MC-III had lower fusion rate and PROs in symptoms and pain compared to those with other subtypes

MC—Modic change; MC-I—type I Modic change; MC-II—type II Modic change; MC-III—type III Modic change; NRS—numerical rating scale; ODI—Oswestry Disability Index; MCID—minimal clinically important difference; TLIF—transforaminal lumbar interbody fusion; VAS—visual analog scale; PLIF—posterior lumbar interbody fusion; SSM—spinal stenosis measure; EQ-5D—EuroQol-5D; OLIF—oblique lateral interbody fusion; SF-12—12-item short form survey; PROs—patient reported outcomes; JOA—Japanese Orthopedic Association; TF-PELD—percutaneous endoscopic lumbar discectomy via a transforaminal approach; LDH—lumbar disc herniation; DDD—degenerative disc disease; PLF—posterior lumbar fusion; ALIF—anterior lumbar interbody fusion; LBP—low back pain; XLIF—extreme lateral interbody fusion, RMDI—Roland–Morris Disability Index; PCS-12—physical component score of the 12-item short form survey.

## Data Availability

All data is contained within the article or appendix.

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
