# Peer review of "Evaluating the Impact of Modic Changes on Operative Treatment in the Cervical and Lumbar Spine: A Systematic Review and Meta-Analysis"

_ijerph, 2022, doi:10.3390/ijerph191610158_

Round 1

Reviewer 1 Report

Lambrechts_Modic Changes_ijerph_2022

I commend the authors on the completion of this manuscript. 

The article includes a comprehensive introduction and background. 

The research question is well defined, being clinically relevant. The presentation defines the research question. 

The research has been carried out in accordance with international recommendations.

I have two comments highlighted below.

General comment

Please, explain the acronyms in the foot of the tables. 

Specific comments

“Appendix A. PRSIMA Checklist.” Please correct to PRISMA.

Author Response

Thank you for your review of our manuscript.

1. Please, explain the acronyms in the foot of the tables. 

We have addressed these concerns with the acronyms in our tables be removing extraneous acronyms and providing a description at the footnotes of all tables and figures.

2. Specific comments

“Appendix A. PRSIMA Checklist.” Please correct to PRISMA.

Thank you for identifying this error. The spelling has been corrected to "PRISMA".

Reviewer 2 Report

I think it is an excellent job. Methodologically, I have no objections. I believe that the review carried out by the authors is exhaustive and sufficient. 

However, it is a subject that has been studied repeatedly, and I believe that the authors should justify more the need for such a review. In other words, what is the contribution of this work? What does it tell us new? Why should we re-evaluate this topic?

Author Response

This study adds the only meta-analysis on the operative treatment of spinal disease in patients with Modic changes to the literature. The literature on Modic changes has experienced significant growth over the past few years. The previous systematic review of lumbar spine surgery outcomes among patients with Modic changes (citation 19) did not draw conclusions regarding lumbar fusion surgery. Several new publications in the interim have contributed significantly to the quality of our review which discusses operative treatment in both the cervical and lumbar spine.

Moreover, previous systematic reviews could not conduct a meta-analysis on the available data while our meta-analysis addresses the impact of Modic changes on patient-reported outcomes following lumbar surgery and suggests a direction for future research. This systematic review and meta-analysis is warranted as previous reviews were limited by the number and quality of studies they were reviewing.